# Small, solubilized platinum nanocrystals consist of an ordered core surrounded by mobile surface atoms

Henry Wietfeldt [1], Rubén Meana-Pañeda[1], Chiara Machello[2], Cyril F. Reboul [1], Cong T. S. Van[1], Sungin Kim [3,4], Junyoung Heo [3,4], Byung Hyo Kim[5], Sungsu Kang[3,4], Peter Ercius [6], Jungwon Park [3,4,7,8✉] & Hans Elmlund [1✉]

In situ structures of Platinum (Pt) nanoparticles (NPs) can be determined with graphene liquid cell transmission electron microscopy. Atomic-scale three-dimensional structural information about their physiochemical properties in solution is critical for understanding their chemical function. We here analyze eight atomic-resolution maps of small (<3 nm) colloidal Pt NPs. Their structures are composed of an ordered crystalline core surrounded by surface atoms with comparatively high mobility. 3D reconstructions calculated from cumulative doses of 8500 and 17,000 electrons/pixel, respectively, are characterized in terms of loss of atomic densities and atomic displacements. Less than 5% of the total number of atoms are lost due to dissolution or knock-on damage in five of the structures analyzed, whereas 10–16% are lost in the remaining three. Less than 5% of the atomic positions are displaced due to the increased electron irradiation in all structures. The surface dynamics will play a critical role in the diverse catalytic function of Pt NPs and must be considered in efforts to model Pt NP function computationally.

[1] Center for Structural Biology, Center for Cancer Research, National Cancer Institute, Frederick, MD 21702, USA. [2] Department of Biochemistry and Molecular Biology, Biomedicine Discovery Institute, Monash University, Melbourne, VIC, Australia. [3] Center for Nanoparticle Research, Institute for Basic Science (IBS), Seoul 08826, Republic of Korea. [4] School of Chemical and Biological Engineering, Institute of Chemical Process, Seoul National University, Seoul 08826, Republic of Korea. [5] Department of Material Science and Engineering, Soongsil University, Seoul 06978, Republic of Korea. [6] National Center for Electron Microscopy, Molecular Foundry, Lawrence Berkeley National Laboratory, Berkeley, CA 94720, USA. [7] Institute of Engineering Research, College of Engineering, Seoul National University, Seoul 08826, Republic of Korea. [8] Advanced Institute of Convergence Technology, Seoul National University, Suwon-si, Gyeonggi-do 16229, Republic of Korea. ✉email: jungwonpark@snu.ac.kr; hans.elmlund@nih.gov

Colloidal nanoparticles (NPs) are inorganic clusters of hundreds to thousands of atoms, which assemble and function in solution[1,2]. They exist in single- and poly-crystalline forms and often assemble into structures with distinct domains[3]. The physiochemical properties of an NP are controlled by its exact atomic structure and composition[4,5]. The exposed surface structure plays a critical role in reactions[6] that have applications in an ever-expanding range of areas, from electronics to catalysis and biological sensors[7]. For example, single Pt atoms behave as spectators in carbon monoxide oxidation and water-gas shift at low temperatures, whereas Pt NPs show distinct catalytic activity[8]. Exposed surfaces, defects, dislocations, and quantum effects are dominant in NPs of finite size[9]. Small Pt NPs (<3 nm in diameter), such as those analyzed here, have a complex surface structure with a large fraction of edge and corner atoms[6,10,11]. These sites often contribute to specific catalytic activities[6,8,12,13]. Furthermore, the passivating organic ligands influence the surface structure[14,15] and are therefore important determinants for reactivity.

We previously introduced Structure Identification of Nanoparticles by Graphene Liquid cell EM (GLC-EM) (SINGLE)—an open source software package for analysis of GLC-EM time-series of individual NPs tumbling in solution[16]. SINGLE has been used to reconstruct many atomic-resolution NP maps that have given a multitude of structural insights, including characterization of lattice deformations and strain[17] as well as analysis of coordination numbers (CNs) to understand the origin of the catalytic performance of Pt catalysts[6]. A critical goal in the field is to identify the variety of NP structures in solution, to understand their domain organization, dynamics, composition, lattice variations, defects, and how these structural features relate to their chemical function both at the single-particle and solution ensemble level. It is especially important to analyze NP structures of different sizes to understand how the structures change when the ratio of surface atoms to the total number of atoms increases and the average degree of coordination decreases. Understanding how the structures change as they grow or shrink may hold answers to long-sought physical models for NP formation, growth, and dissolution.

There are many dynamic effects that influence the image characteristics of the nanocrystal views acquired along the time-series. Often, the liquid cell is thicker in the beginning of the time-series and thins as the liquid evaporates over time. This causes the liquid background signal to be stronger in the beginning, which may obscure atomic contrast and lead to worse image quality. As the cell thins, atomic contrast is enhanced but thinning of the cell typically reduces the rotational freedom of the nanocrystal, leading to poor orientational coverage and worse 3D reconstructions as a result. Another important phenomenon is that the nanocrystals move laterally in the liquid cell and out of the narrow depth of focus of the aberration-corrected TEM, which causes loss of atomic contrast. We previously introduced computational methods that address these challenges[3,16,17]. Furthermore, the nanocrystals may undergo inherent dynamic structural changes due to growth, dissolution, and thermal motion. These effects may occur naturally in solution and cannot be straightforwardly distinguished from effects introduced by the electron beam. Gross time-dependent structural changes can readily be detected already in the 2D analysis step. All the time-series from which 3D density maps were previously calculated were deemed stable under the imaging conditions used through 2D analysis before 3D reconstruction[3,16,17].

Despite the high degree of sophistication and automation of the SINGLE procedures, many improvements remain to be made, particularly in automating the atomic model-building step—the goal being to develop quantitative analysis tools for characterizing atomic-scale properties in an automated fashion. Another reason to improve and automate the structure solution steps in SINGLE is to make the method accessible to inexperienced users and suitable for high-throughput analysis. Automation of the atomic model-building step presents several challenges, including denoising of the reconstructed 3D map, determination of the optimal map threshold (often called sigma value in the structural community, because the cutoff at which the map is rendered in 3D is expressed in number of standard deviations), binary segmentation of the map to identify connected components (atoms), density modification to split atoms that remain connected in areas of lower local resolution, and validation of the built atomic model. Here, we present computational methods addressing all these steps. We also introduce approaches for characterizing the isotropic and anisotropic atomic displacements. Atom mobility is analyzed in the context of other atomic-scale structural features, such as degree of coordination, lattice deformations, strain, and bond length. We knew from our previous work that the solution ensemble of synthesized Pt NPs is heterogenous—each NP has a unique atomic structure[3,16,17]. However, in this study, we reveal that the individual structures are themselves heterogeneous, displaying a strong radial and CN dependence of critical atomic-scale statistics, such as isotropic and anisotropic mean-square atomic displacements, atomic peak intensity, and atomic shape characteristics. Our two main findings are (1) that the individual Pt NPs have an ordered crystalline core surrounded by a comparably mobile and irregular surface structure and (2) that the penetration depth of the effect of the solvent on the surface atom mobility corresponds to the 1.6 outer shells of atoms on average (3.9–5.1 Å) for Pt NPs with a diameter larger than 20 Å. Finally, we conclude that our findings hold true when the cumulative electron dose used to calculate the final 3D reconstructions is increased from 8500 to 17,000 electrons/pixel.

## Results and discussion
**Time-averaged 3D reconstruction.** We imaged eight Pt NPs of varying sizes using GLC-TEM[3,17], and reconstructed a time-average 3D volume for each NP time-series using SINGLE[16]. For each reconstructed 3D volume, we determined the atomic centers using the algorithms described in Materials and Methods. Figure 1 summarizes our methodology. Table 1 provides an overview of the eight NP structures analyzed. We calculated several atomic statistics in an unsupervised manner (see Fig. 1 and Materials and methods), to identify common trends and characterize differences between the structures. Our code developments are available open source as part of the SINGLE software package, available for download at https://github.com/hael/SIMPLE3.0.git. The atoms in an NP move during imaging due to thermal oscillations, causing displacement from their ideal lattice positions. This atomic movement results in smearing of the atomic intensity distributions since the experimental map is a time-average of the NP during imaging. This "smearing" is quantified by isotropic and anisotropic atomic displacements parameters, commonly referred to as B-factors[18–23]. Solvent effects become increasingly important as we move from the core of highly coordinated atoms to the less coordinated surface atoms, which are more susceptible to being affected by interactions with the passivating organic ligands and the interacting solvent molecules. Therefore, a fundamental characteristic of small Pt NPs is that they have an ordered, highly coordinated core and an irregular and potentially mobile surface structure.

We calculated the median CN, nearest neighbor (NN) bond length, radial strain, and core ratio (fraction of atoms with CN = 12) for each NP (see Table 1). NPs with larger diameters tend to have a higher ratio of core to surface atoms. Therefore, the

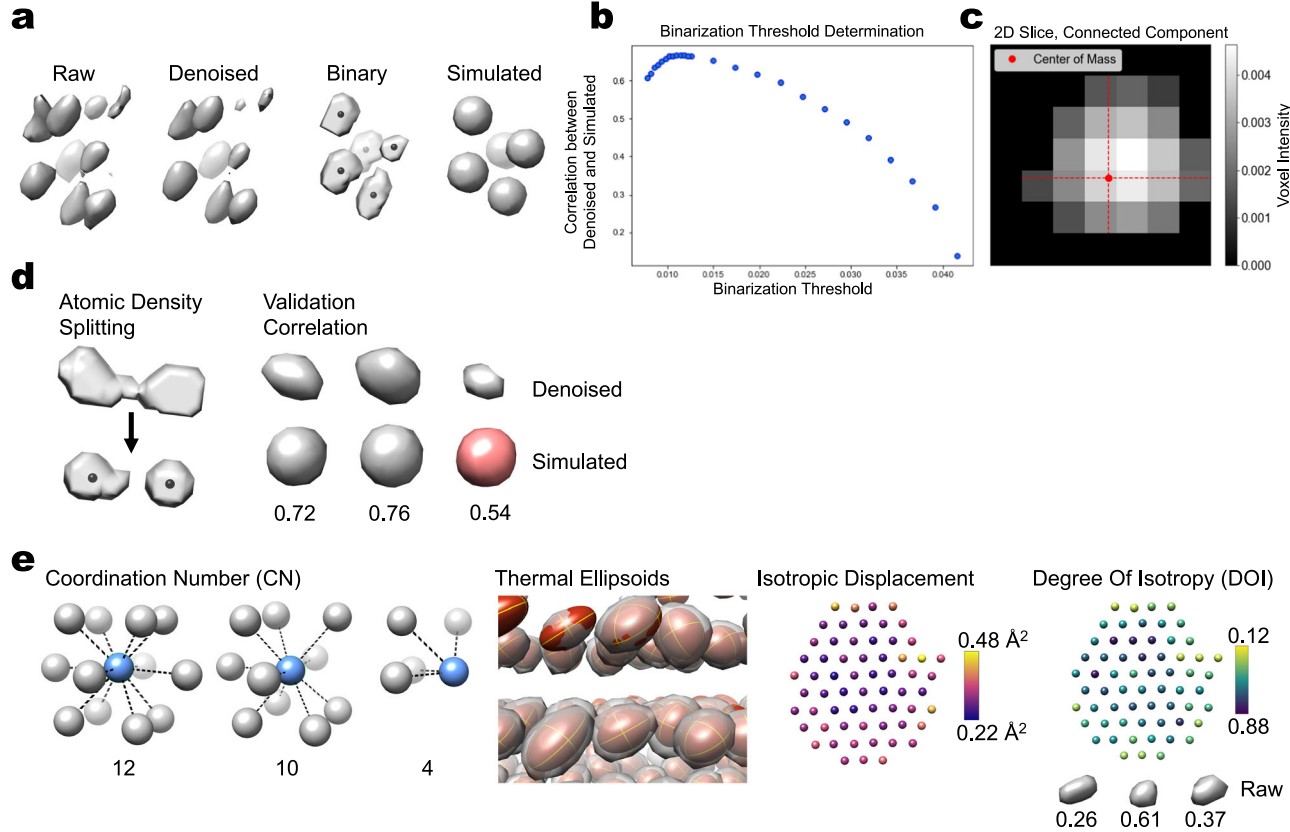

**Fig. 1 Schematic summary of our method for unsupervised atomic-scale analysis of high-resolution GLC-EM density maps (see Materials and methods for details). a–c** Atomic density detection and identification of atomic positions. **a** Denoising of the experimental map (raw vs. denoised), binarization (binary), and map simulation (simulated). **b** Determination of the optimal map binarization threshold through maximization of the correlation between simulated and denoised maps. **c** Atom positions are assigned as the centers of mass of the grayscale voxel values within the connected components. **d** Validation of atomic positions. Atomic densities that are still connected due to limited local resolution are split. Outlier atoms are rejected based on atomic density quality (validation correlation). **e** Unsupervised per-atom statistics calculations include coordination number (CN) and atomic displacement parameters. Thermal ellipsoids representing fit anisotropic displacement parameters. Example central plane of atoms in NP4 shows that atoms close to the surface move more (isotropic displacement) and with a greater degree of isotropy (DOI). Examples of atomic densities with different DOI in NP4 illustrate the heterogeneity of atomic thermal motion (bottom right).

**Table 1 An overview of the NP structures analyzed in this study.**

| Name | Reference | Number of atoms | Diameter (Å) | Median CN | Median NN bond length (Å) | Median radial strain (%) | Core ratio (fraction of atoms with CN = 12) |
|---|---|---|---|---|---|---|---|
| Nanoparticle 1 (NP1) | 16 | 219 | 21.3 | 8.8 | 2.57 | 1.6 | 0.16 (35) |
| NP2 | 16 | 357 | 25.1 | 9.5 | 2.76 | 4.4 | 0.40 (142) |
| NP3 | 17 | 373 | 23.5 | 9.7 | 2.75 | 1.7 | 0.44 (163) |
| NP4 | 17 | 390 | 23.7 | 9.8 | 2.75 | 1.8 | 0.46 (179) |
| NP5 | 17 | 496 | 26.0 | 9.9 | 2.72 | 2.1 | 0.49 (242) |
| NP6 | 17 | 569 | 27.3 | 9.9 | 2.64 | 4.7 | 0.43 (242) |
| NP7 | 16 | 667 | 29.5 | 9.9 | 2.59 | 1.4 | 0.41 (275) |
| NP8 | 17 | 710 | 30.0 | 10.1 | 2.64 | 1.8 | 0.51 (365) |

The NP diameter is defined as the maximum distance between any two atoms within the NP. The core ratio is computed as the number of atoms with CN = 12 by the total number of atoms.

median CN increased with NP diameter. There was no clear trend between NP size and median bond length. Radial strain was 1.4–4.7% on average, agreeing with previous findings that ligand-protected metal NPs display a small expansion with respect to an ideal Pt lattice[17,24,25]. Interestingly, there was no clear trend between median radial strain and diameter. We assumed that larger NPs would have a lower median radial strain, since a larger percentage of atoms belong to the core and presumably would not

deviate significantly from an ideal crystal lattice. Instead, radial strain was roughly uniform in the NP size range investigated here.

**Beam-induced dynamic structural effects**. Core atoms tended to have CN = 12 and surface atoms lower coordination, as shown in Fig. 2a. NP1 is the smallest of the NPs analyzed here and nearly all atoms had CN < 12 with significant disorder. We wanted to assess the possible beam-induced effects on lowly coordinated

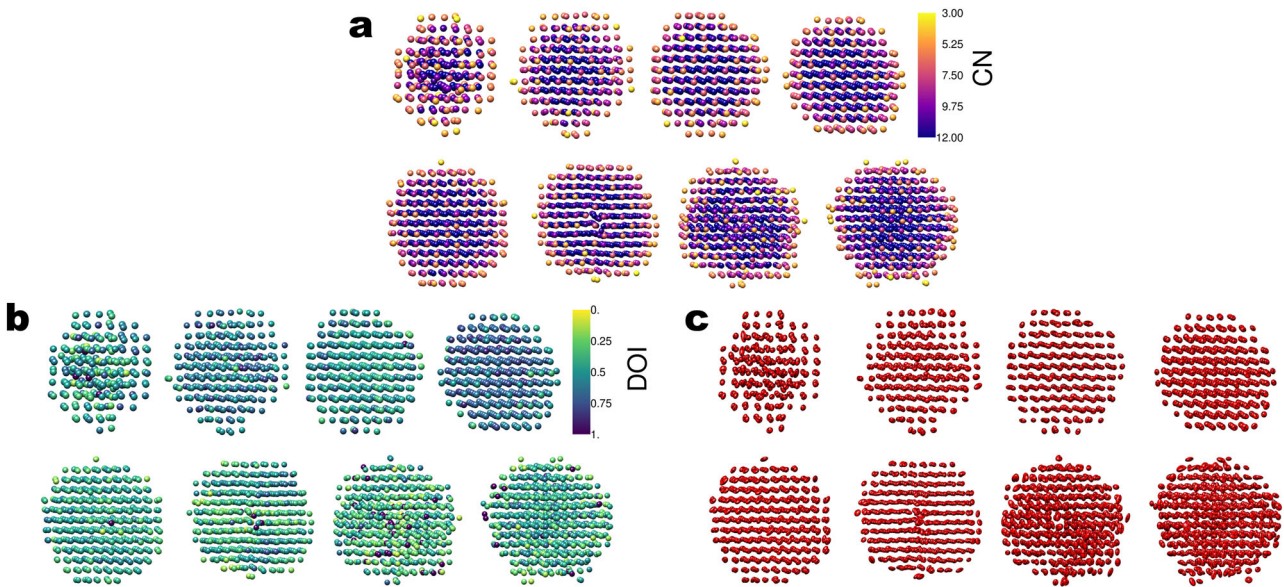

**Fig. 2 Atomic models of NPs 1–8 (ordered left to right, top to bottom in each subfigure). a** Atoms colored by coordination number (CN). **b** Atoms colored by degree of isotropy (DOI). **c** Atoms represented with "thermal ellipsoids," in which semiaxis lengths are square roots of $U_{maj}$, $U_{med}$, $U_{min}$ with directions corresponding to the respective eigenvectors of the displacement parameters matrix **U** (see Materials and methods for definition). The sizes of atoms in (**a**) and (**b**) are irrelevant.

surface atoms. Given the dose regime used to collect the data[16], it is likely that the passivating organic ligands that cover the surface of the nanocrystals are damaged in the first few frames and that nanocrystals with a damaged passivation layer are imaged for the remainder of the time-series. To what degree radiation damage of the organic material affects the configuration of the metallic surface atoms would have to be addressed with low-dose TEM[26–28]. This is beyond the scope of this study. However, to test the effect of changing the cumulative electron dose, we split the time-series for all eight data sets analyzed into halves and produced 3D reconstructions from the first and second halves (part1 and part2). The cumulative electron dose was 8500 electrons/pixel and 17,000 electrons/pixel for part1 and part2, respectively. Atomic models were built automatically and independently for the two parts of each time-series, using the algorithms described in Materials and Methods. Figure 3 summarizes the results. Minor dynamic effects were observed, including dissolution of surface atoms, and small changes in atomic positions of common atoms between the two parts. It is impossible to tell whether these dynamic phenomena would occur naturally in solution or are a consequence of interactions with the electron beam. Comparing the atomic models obtained for part2 with those obtained for part1, less than 5% of the total number of atoms were lost for all NPs except for NP1, NP7, and NP8 which lost 12%, 16%, and 10%, respectively. Whether or not the observed compositional differences are real is unclear, because surface atoms exhibit larger thermal vibrations than core atoms, which lowers their effective signal-to-noise ratio (SNR). Hence, when the time-series is split, the SNR of surface atom densities are weakened more than core atom densities, which may put them below the threshold for detection. Next, we analyzed the atomic positions in the two parts and defined common atomic positions as those that had not changed by a distance larger than 0.8 times the atomic diameter of Pt (1.76 Å, which is equivalent to 0.45 times the lattice parameter of bulk Pt). All the NPs had 95% or more common atomic positions. The number of atomic positions that differed in part2 versus part1 increased with increasing size of the NPs and predominantly involved surface atoms, as expected.

**Solvent penetration depth analysis**. Next, we plotted CN versus radial depth $R_{NP} - R$, where $R$ is the distance between an atom and the NP center and $R_{NP}$ is the NP radius, defined as half the maximum distance between any two atoms in the NP. Figure 4a shows these plots for all eight NPs. We averaged the CN of every five consecutive atoms after ordering them according to radial depth to generate a continuous average radial CN statistic. For all NPs except for NP1, CN increased linearly with radial depth until it plateaued at CN ≈ 12 beginning at some depth $\delta$ into the NP. These transition points were not smooth but indicated a sudden transition between the surface and core. To determine $\delta$ for each NP, the plots in Fig. 4 were fit with a continuous piecewise linear-constant function.

$$\mathrm{CN} = \begin{cases} m\left(R_{NP} - R - \delta\right) + c \text{ if } R_{NP} - R \leq \delta \\ c \text{ if } R_{NP} - R > \delta \end{cases}$$

where $m$, $c$, and $\delta$ are fitting parameters. The parameter $c$ was restricted to the interval [10, 12] to reflect that atoms in an ideal lattice have CN = 12. $\delta$ was similar for all NPs, ranging from ~4 Å (NP4) to ~5 Å (NP2). The fact that $\delta$ was similar for all NPs suggests that crystal abnormalities resulting from NP surface interactions with ligands, the solvent, and the electron beam have a "penetration depth" $\delta$, beyond which the NP approximates an ideal FCC lattice. It is at this depth $\delta$ that a transition occurs from a lowly coordinated, irregular surface to a highly coordinated crystalline core. Given that the outermost surface atom of an NP has a depth of 0, assuming overall spherical NP shape, and a spacing between the atoms of approximately $\frac{\bar{a}}{\sqrt{2}} \approx 2.8$Å, then based on these fits roughly the 1.6 outer shells of atoms constitute the disordered "surface" region. To assess beam-induced effects on the estimated penetration depth ($\delta$), we repeated these analyzes for the 3D reconstructions generated from the split time-series (part1 and part2). In all cases, the estimated $\delta$ for part1 (red data points in Fig. 4) and part2 (green data points in Fig. 4) clustered closely together with the $\delta$ estimated for the merged time-series (black data points in Fig. 4). The three $\delta$ values for NP1—the smallest NP—showed the greatest variation, because binary splitting of the NP1 time-series caused the largest decrease

**Fig. 3 Atomic models generated from the entire time-series (merged), the first half (part1), and the second half (part2).** The number of atoms is indicated below each structure. The percentage of common atoms between part1 and part2 is indicated (common (%)). Atoms that were displaced more than 1.76 Å in part2, when compared to part1, are shown in the column labeled different.

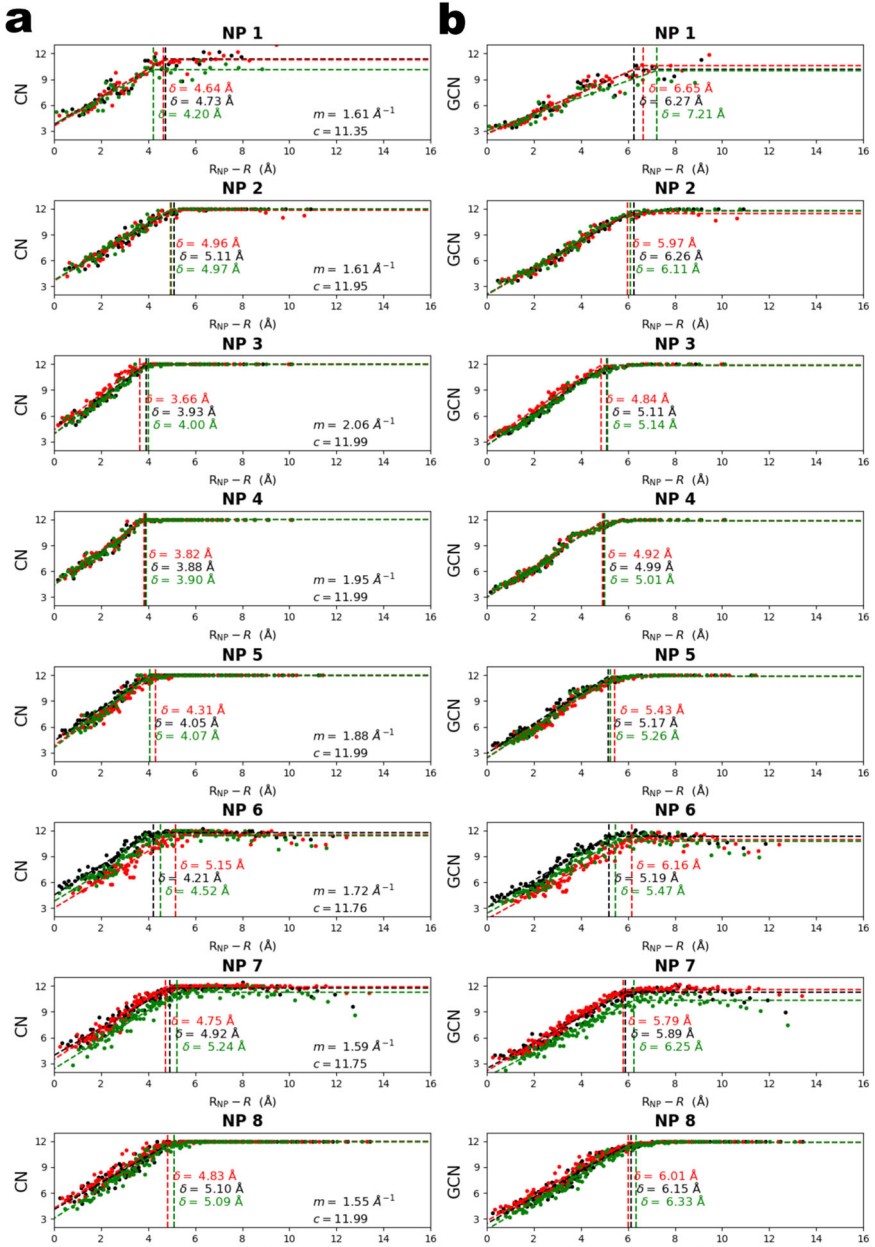

**Fig. 4 Solvent penetration depth analysis. a** Coordination number (CN) and **b** generalized coordination number (GCN) versus radial depth for each of the eight NPs. Each point is an average of five atoms with similar radial positions. Dashed lines show fits using Eq. 1. Vertical dashed lines indicate the penetration depth $\delta$ estimated from the fit. Red, green, and black data points correspond to part1, part2, and merged, respectively.

in effective SNR of the part1 and part2 3D reconstructions due to NP1 having the least scattering mass of all NPs.

**Coordination-dependent atomic statistics**. We plotted the above-described atomic statistics versus CN (Fig. 5) to further illustrate this transition between surface and core atoms. The strong trends in Fig. 5 show that an atom's CN is a determinant of its local structural properties within the NP, as expected. Figure 5a, b show that atoms with lower CNs had greater thermal motion than atoms with higher CNs. The more an atom moves during imaging, the broader its intensity distribution. When an atom's average displacement increases, its peak intensity decreases. We defined the per-atom DOI (Degree Of Isotropy) statistic such that an atom has a DOI of one if its displacements are purely

isotropic and zero if they are confined to a plane or a line (see Materials and Methods). Figure 5c shows that the DOI increased with CN, indicating that atomic thermal motion was more anisotropic near that surface than in the core. We observed large variation in DOI between NPs, although there was no correlation between DOI and NP size. The validation correlation $\in [-1,1]$ describes how well an experimental atomic density agrees with the corresponding simulated density, assuming purely isotropic displacement (see Materials and Methods). The validation correlation increased with CN, as illustrated in Fig. 5d, indicating that atoms with CN = 12 better match the expected intensity distributions from an ideal FCC lattice of Pt atoms undergoing purely isotropic motion. Combining these dependencies on CN with the observed sharp transition between a lowly coordinated surface region and highly coordinated core, it is evident that

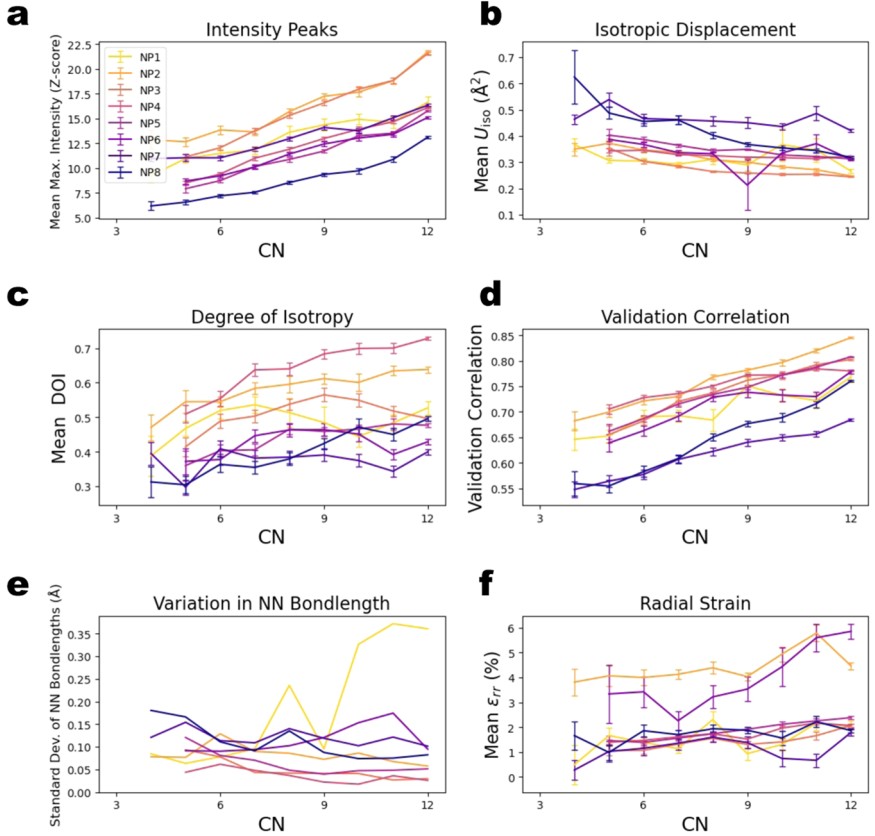

**Fig. 5 Various atomic statistics binned by coordination number (CN) for the eight NPs.** Only bins with ≥5 atoms are shown. Error bars are calculated as $S/\sqrt{N}$ where $S$ is the standard deviation of the relevant statistic within the bin and $N$ is the number of atoms in the bin. **a** Peak intensity $z$-score. **b** Isotropic displacement. **c** Degree of isotropy. **d** Validation correlation. **e** Variation in nearest neighbor (NN) bond length without error bars. **f** Radial strain.

atoms in the surface region move more than atoms in the core region and with greater anisotropy. Furthermore, the nature of change in CN with radial depth at $r_0 - r = \delta$ suggests that the transition between the surface region and core region within these PVP-ligated Pt NPs is abrupt. Bond length (Fig. 5e) and radial strain (Fig. 5f) are similar between the "core" and the "surface" regions, whereas the key to the explanation of this dichotomous phenomenon lies in the atomic displacements.

An atom's properties do not only depend on its own CN but also the CNs of its neighbors and the CNs of its neighbors' neighbors and so forth. Therefore, we also examined the penetration depth when calculated using the generalized coordination number (GCN) (see Fig. 4b and Materials and methods). The penetration depths for GCN were similar across the NPs, ranging from ~5 to ~6 Å. These penetration depths were larger than the penetration depths calculated with CN because atoms at the outer edge of the core region are neighbors to lower coordinated atoms at the interior edge of the surface region. Considering that atomic spacings were ~2.8 Å, we expected penetration depths calculated with GCN to be between 0 and 2.8 Å greater than those calculated with CN, which was generally what we found. Likewise, atoms at the inner edge of the surface region are neighbors to atoms in the core region which generally have CNs of 12, so GCN vs. radial depth curves didn't display as sharp a transition as the CN curves.

**A dynamic structural model for small Pt NPs.** Plots of isotropic displacement, maximum intensity, DOI, and validation correlation versus radial depth were fit with piecewise linear-constant functions (Eq. 1, with removal of the constraint on the parameter $c$ and necessary changes to units) for all eight NPs (Fig. 6 and

Supplementary Fig. 1) to confirm that there was an abrupt change in atomic thermal motion between surface atoms and core atoms. Penetration depths identified with these statistics showed greater variation than penetration depths estimated based on CN between the eight NPs. Nonetheless, the discontinuity in the derivative of the curves was generally between one to two lattice spacings (2.8–5.6 Å), indicating a transition between surface and core regions. In the cases where the penetration depth measured using a particular atomic statistic was less than 3 Å, the innermost atoms displayed surface-like properties, illustrating that the model of an ordered core and disordered surface is a simplified one. For example, for NPs 4–8, the isotropic thermal motion $U_{iso}$ increased for the innermost atoms, suggesting that they behaved more like "surface" atoms than "core" atoms, even though they were buried deep in the core. Hence, there must be properties not described by CN or GCN that cause a return to "surface-like" properties near NP centers. While the cause of the increasing thermal motion at the NP centers is unclear, it affected only the most central atoms in NPs 4–8 and wasn't present in NP2 and NP3. Thus, the binary model of a mobile surface region and an ordered, crystalline core still serves as a reasonable model for these small Pt NPs. Importantly, some statistics displayed a smooth transition between core and surface regions, as shown in Fig. 6, appearing more logistic than piecewise, so a hard, discontinuous cutoff between surface and core regions approximates a more complex transition.

## Conclusion

We introduce unsupervised methods for NP density map thresholding and atomic position determination, providing the basis for automated analysis of atomic intensity, shape, and

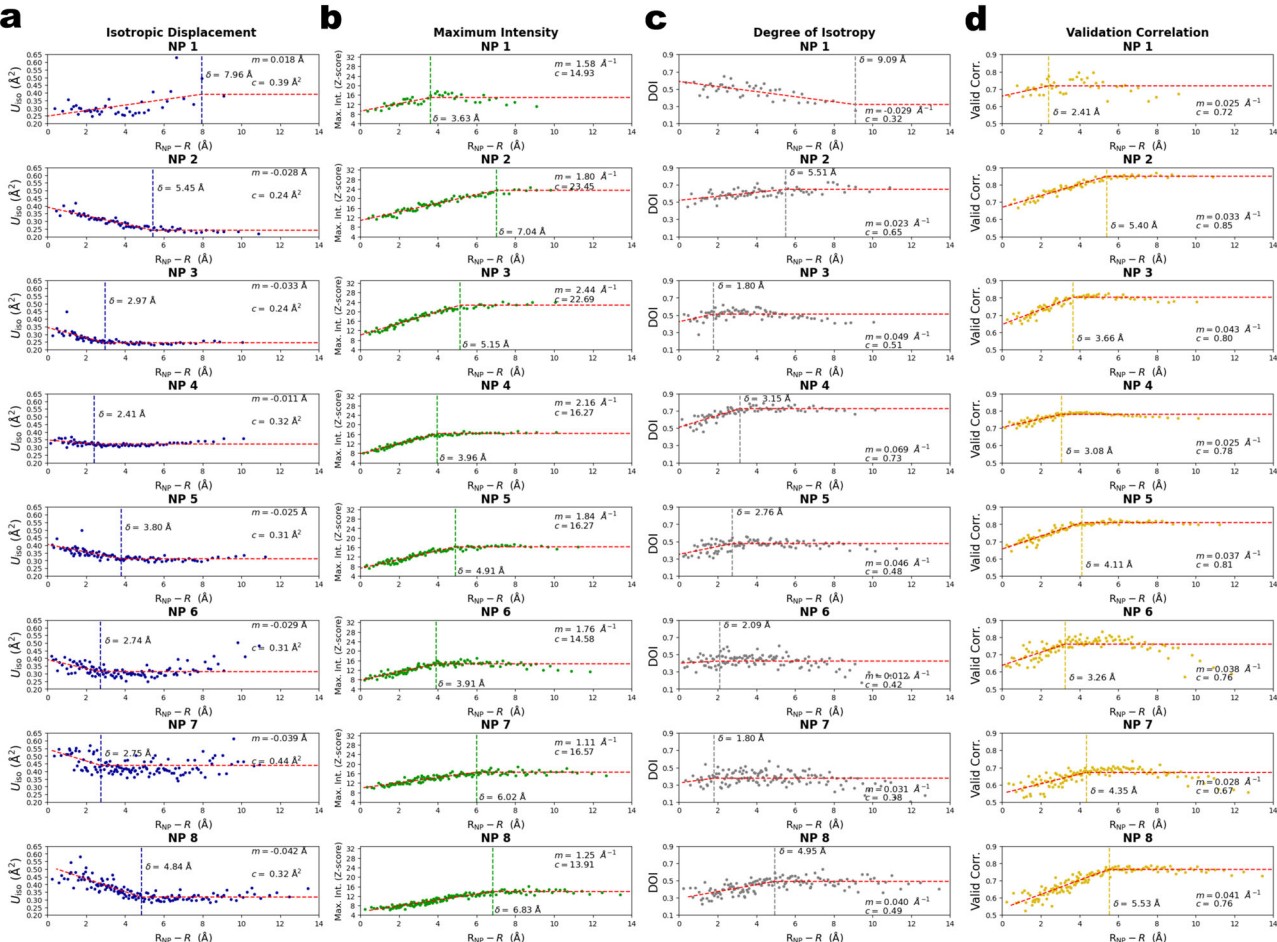

**Fig. 6 Penetration depth plots for various atomic statistics. a** Isotropic displacement, **b** maximum intensity, **c** degree of isotropy (DOI), and **d** validation correlation versus radial depth for each of the eight NPs. Each point is an average of five atoms with similar radial positions. Red dashed lines show fits using Eq. 1. Vertical dashed lines indicate the penetration depth $\delta$ estimated from the fit. Best fit values for parameters $m$ and $c$ are also shown.

displacement with respect to the distance from the surface of the NPs. We conclude that metrics based on deviation from ideal FCC crystallinity are not sufficient to explain the structural characteristics of these small (<3 nm) Pt NPs. Rather, the key to the explanation of the observed dichotomous phenomenon—an ordered core surrounded by comparably mobile surface atoms—lies in the atomic displacements. The intrinsic surface dynamics of Pt NPs revealed in this study will be critical for improving the accuracy of property prediction through computational chemistry and must be considered when modeling these or similar systems.

An interesting future development would be to collect data with different dose rates of homogenous solution ensembles of so-called "magic number" nanocrystals[29] that have had their structure determined by "classical" single-particle 3D averaging in conjunction with TEM imaging in dry conditions[9,30]. This is the only scenario that we can conceive of where dose rate-dependent phenomena could be reliably characterized. The individual nanocrystals imaged with GLC-EM are never identical between any two frames of the time-series due to dynamic phenomena at the atomic scale. The exercise of splitting the time-series into contiguous segments is therefore a tradeoff between the averaging that is needed to improve the SNR of the reconstructed 3D densities and the temporal resolution at which dynamic effects can be reliably detected. Atom detection and extraction of atomic shape statistics critically depend on the SNR of the reconstructed 3D density. We are confident that the nanocrystals analyzed here were sufficiently stable during data acquisition that averaging

across the entire time-series is meaningful for most atoms. Dynamic surface effects were observed, but whether they represent real structural differences or are a consequence of the lower effective SNR in the split time-series is unclear. It is likely that radiation damage of the organic ligands that cover the nanocrystal surface occurs already at a very low electron dose (5–10 e⁻/Å²), as for proteins, and that we are in fact imaging NPs with a damaged passivation layer. Improved data acquisition schemes, possibly under cryogenic conditions[31], applied to the appropriate NP systems in conjunction with robust quantitative image processing are needed in the future to further characterize these dynamic phenomena.

Our unsupervised atomic model-building method will open for development of novel approaches for 3D refinement, where the automated model-building step is combined with the iterative 3D orientation refinement, as has been done for more than 20 years in the field of protein crystallography[32]. This may prove critical for our ability to reconstruct smaller NPs, where the diffuse scattering due to the liquid background constitutes a significant part of the total signal, resulting in biased 3D reconstructions. Furthermore, increased robustness of the 3D refinement through coupling it with automated atomic model building and simulation will be critical for our ability to resolve different structural states along the time-series, when the NP undergoes dynamic phenomena, such as growth or shrinkage due to dissolution. We believe that the foundation laid out in this study will provide the necessary framework for future characterization of the structural

variations that occur along the time-series and extraction of time-dependent atomic displacement parameters.

## Methods

Pt NPs were synthesized, graphene liquid cells prepared, TEM images acquired, and 3D reconstructions generated as previously described[16].

**Method summary**. Our method for unsupervised atomic-scale analysis of high-resolution GLC-EM density maps is summarized below (also see Fig. 1).

- Automated atomic model building

    - Atomic density detection
        - Denoising of the experimental map
        - Determination of the optimal map binarization threshold
        - Binarization of the map
        - Identification of connected components (atoms)

    - Unsupervised identification of atomic positions
        - Atom positions are assigned as the centers of mass of the grayscale voxel values within the connected components

    - Validation of atomic positions
        - Split atomic densities that are still connected due to insufficient local resolution and re-determine atomic positions for modified density segments
        - Remove outlier atom positions based on atomic density quality and degree of coordination

- Unsupervised per-atom statistics calculation

    - Atomic position statistics
        - Standard coordination number (CN)
        - Generalized coordination number (GCN)
        - Nearest neighbor (NN) bond length
        - Radial strain

    - Atomic shape statistics
        - Maximum intensity
        - Validation correlation
        - Isotropic atomic displacement
        - Anisotropic atomic displacement
        - Degree of isotropy (DOI)

### Automatic atomic model building

*Denoising of the experimental map*. The map reconstructed with SINGLE[16] was too noisy to analyze directly with binary segmentation techniques. We therefore substituted map $M$ by a map $C$ that was the result of the correlation between map $M$ and a map representing a single simulated atomic density of the element that the NP is composed of, thus utilizing well-known properties of the Fourier transform[33] in conjunction with a priori knowledge about the elemental composition of the NP. We used the denoised map $C$ in all following calculations unless otherwise stated.

*Unsupervised identification of atomic positions*. We applied the binary segmentation techniques described in ref. [34] to simplify the voxel-based representation of map $C$. The map was binarized, producing map $B$, and connected components (atoms) identified. We defined each atom center as the center of mass[35] of the gray-level voxel intensities in map $M$ across its corresponding connected component in map $B$. Selecting the correct threshold $T$ for determining which density is considered foreground (atoms) and which is considered background (noise and solvent) in map $C$ is a nontrivial but important problem. Determining $T$ is not only important for understanding how many atoms a given NP is composed of, but inappropriate thresholding can lead to incorrect interpretation of the surface structure. In computer vision and image processing, Otsu's method is used to perform automatic segmentation of an image into foreground and background pixels[36,37]. However, Otsu's method is sensitive to noise and for the NP maps analyzed here it provided a threshold that included too much foreground density. We therefore developed a method for automatic determination of $T$, resulting in a binary representation $B$ of map $C$ (Fig. 1a). An initial threshold $T_0$ was estimated using Otsu's method and refined as follows. We produced the gray-level histogram of map $C$, and generated 15 uniform thresholds in the range from $T_0$ to the maximum value of map $C$. For each of the 15 threshold values, we binarized the input volume, identified connected components, and determined atomic positions, as described above. Next, we simulated 15 maps based on the atomic coordinates, as described in refs. [16,20,38], and calculated the volume-to-volume correlations between map $C$ and the simulated maps. We subjected the discrete threshold $T$ that maximized the correlation to further refinement using finer $T$ sampling. After binary segmentation, a small number of the atomic densities still represented two or more atoms due to limited local resolution.

*Validation of atomic positions*. At this stage, connected atoms were still present in the binary map $B$, which were considered as one connected component (therefore as one atom) instead of two or more. The center of the gray-level intensities of such a connected component falls in the proximity of the fragment of density that connects the two atoms. Atom centers should coincide with peaks in map $C$. Therefore, correctly identified atom centers would be "close" to peaks in $C$, whereas erroneously identified atom centers would be "far" from peaks in $C$. Therefore, we extracted the value of map $C$ at the suggested atom center. If the center value was more than one voxel away from the maximum of map $C$ within the considered connected component or the longest voxel to center distance within the connected component was larger than 1.5 times the theoretical atomic radius, we split the connected component (Fig. 1b) and defined new atom centers. The algorithm is outlined below.

Initialization: select the first new center $C_0$ as the maximum value of map $C$ across the connected component. Step $i = 1,2…$ of the algorithm:

1. Draw one atom $A_{i-1}$ centered in $C_{i-1}$ with radius equal to the theoretical radius.
2. Intersect $A_{i-1}$ with the connected component to create a new atom $A_{new}$.
3. Select another center $C_i$ as the maximum value of the connected component in map $C$ while ignoring $A_{new}$.
4. If the distance between $C_i$ and $C_{i-1}$ is less than twice the theoretical atomic radius stop. Otherwise, go to step 1.

We refined the binary map further by identifying and eliminating outlier atoms. Comparison of the simulated map $S$ with the experimental map $M$ allowed calculation of per-atom correlation values, using Pearson's correlation coefficient, calculated over the voxels extracted from the corresponding atomic densities of the respective maps within a sphere with 1.5 times the radius of the expected atomic radius, centered on the atomic coordinate. We discarded atoms with correlation values more than two standard deviations away from the mean on the left-hand side of the distribution. The main reasons for the

discrepancy between the simulated atomic densities and the experimentally derived ones (not considering the various sources of noise) are isotropic and anisotropic atomic displacements, commonly referred to as B-factors[18–23]. We calculated per-atom CNs, as previously described[6], and discarded atoms with CN of less than three. The user of SINGLE can turn off this final thresholding or input any desired CN threshold value.

**Unsupervised per-atom statistics calculation.** In addition to automatic atomic position determination, we developed unsupervised methods for the calculation of atomic statistics, given the experimental map $M$ and the binary connected component map $B$. In this section, we describe the atomic statistics that can be used to analyze structural differences within and between NPs as well as their algorithmic implementation in SINGLE[16]. We used these atomic statistics to characterize the structural properties of eight small (<3 nm) Pt NPs synthesized in the same batch (see Results).

### Atomic positions statistics
*Standard coordination number (CN).* The standard CN of an atom is defined as its number of adjacent neighboring atoms. An atom in an ideal monatomic face-centered cubic (FCC) lattice has a CN = 12. Surface atoms have a CN < 12 that depends on the surface geometry. Interior atoms can also have a CN that differs from 12 if lattice distortions cause the number of adjacent atoms to increase or decrease. We calculated an atom's CN by first fitting the NP's atomic map with a crystal lattice using least squares regression. Next, the atom's neighborhood was defined as a sphere of radius $d = \frac{\bar{a}}{2}(1 + \frac{1}{\sqrt{2}})$ centered about the atom, where $\bar{a}$ is the arithmetic mean of the three lattice parameters and the numerical factor is explained by the geometry of an FCC lattice. We defined an atom's CN as the number of atoms within this spherical neighborhood of radius $d$.

*Generalized coordination number (GCN).* Surface atoms with lower CNs are more likely to form bonds with external species than surface atoms with greater CNs. A better predictor of a surface atom's tendency to form bonds is its GCN—an extension of CN that includes information about the CNs of the atom's NNs. Specifically, the GCN of atom $i$ with $j$ NNs is defined as

$$\text{GCN}_i = \sum_{k=1}^{j} \frac{CN_k}{CN_{\max}}$$

where $CN_k$ is the CN of the $k$th NN of atom $i$ and $CN_{\max}$ is the maximum CN of the $j$ neighbors. GCN is a better predictor of surface catalytic activity than CN[6,12,13,39,40] because a surface atom's tendency to form bonds with external species depends not only on its current number of bonds but also on its surrounding surface geometry.

*Nearest neighbor (NN) bond length.* SINGLE calculates the NN bond length of each atom by finding the minimum distance from the atom of interest to any other atom in the NP.

*Radial strain.* The atomic positions of small Pt NPs do not exactly match an ideal FCC lattice. We calculated the radial strain of each atom to quantify the degree to which an atom deviates from an ideal lattice structure. If $u_R$ is the radial displacement from the ideal lattice structure, then the radial strain $\epsilon_{RR}$ is defined as $\epsilon_{RR} = \frac{\partial u_R}{\partial R}$, where $R$ is the distance from the NP center.

### Atomic shape statistics
*Maximum intensity.* SINGLE reports the maximum intensity of each atom expressed as the number of standard deviations that

the peak grayscale voxel value is away from the mean grayscale voxel value within the connected component (i.e., the region occupied by the atom) in the density map.

*Validation correlation.* In one of the final steps of atomic position determination in SINGLE, we compared detected atoms to their simulated counterparts. For each detected atom, we calculated Pearson's correlation coefficient between the voxel values of the experimental map and the simulated map within a cubic window of side length $3r_{th}$, centered about the detected atom, where $r_{th}$ is the theoretical atomic radius. We termed this correlation coefficient the validation correlation of the atom.

*Isotropic atomic displacement.* The atoms in an NP move during imaging due to thermal oscillations, causing displacement from their ideal lattice positions. This atomic movement results in smearing of the atomic intensity distributions since the experimental map $M$ is a time-average of the NP during imaging. Mathematically, this smearing can be represented by the convolution of the electric charge density with the Debye–Waller Factor[19,21,23]. If $\mathbf{r} = \mathbf{r'} - \mathbf{r_0}$, where in an arbitrary coordinate system $\mathbf{r'}$ is the position vector of the point in question and $\mathbf{r_0}$ is the position vector of the atomic center, then the observed atomic intensity distribution $g(\mathbf{r})$ of a given atom can be expressed as

$$g(\mathbf{r}) = \rho(\mathbf{r}) * T(\mathbf{r})$$

where $\rho(\mathbf{r})$ is the atom's charge density, $T(r)$ is the atom's real-space distribution of the Debye–Waller Factor, and the $*$ operator denotes convolution[19,23]. Assuming harmonicity of the atomic interactions, the Debye–Waller Factor is defined in reciprocal space as

$$T(\boldsymbol{q}) = \exp[-2\pi^2 \langle (\boldsymbol{q} \cdot \boldsymbol{u})^2 \rangle]$$

where $\mathbf{q}$ is the scattering vector and $\mathbf{u}$ is the displacement of the given atomic center from equilibrium[23]. Thus, $G(q)$—the Fourier transform of $g(r)$—can be written as

$$G(\boldsymbol{q}) = f(\boldsymbol{q}) \exp[-2\pi^2 \langle (\boldsymbol{q} \cdot \boldsymbol{u})^2 \rangle]$$

where $f(\mathbf{q})$ is the atomic form factor—the Fourier transform of $\rho(\mathbf{r})$. For isotropic atomic displacements, $\langle (\mathbf{q} \cdot \mathbf{u})^2 \rangle = \frac{1}{3} q^2 \langle u^2 \rangle$, where $\langle u^2 \rangle$, is the mean-square displacement of the atom, and so

$$G(q) = f(q) \exp\left[-2\pi^2 \frac{q^2 \langle u^2 \rangle}{3}\right]$$

The atomic form factor $f(q)$ is often modeled as the sum of five isotropic Gaussians[38]; however, considering that the observed $G_j(q)$ is approximately Gaussian, $f(q)$ must also be approximately Gaussian since the product of two Gaussians is Gaussian. Thus, approximating $G(q) \approx G(0)e^{-2\pi^2 \sigma_g^2 q^2}$ and $f(q) \approx f(0)e^{-2\pi^2 \sigma_\rho^2 q^2}$, where $\sigma_g$ and $\sigma_\rho$ are the real-space isotropic standard deviations of the observed intensity distribution and charge density of the given atom, respectively, we have

$$G(q) \exp[-2\pi^2 \sigma_g^2 q^2] = f(0) \exp\left[-2\pi^2 \left(\sigma_\rho^2 + \frac{\langle u^2 \rangle}{3}\right) q^2\right]$$

which results in the variance relation
$$\sigma_g^2 = \sigma_\rho^2 + \frac{\langle u^2 \rangle}{3}$$

We can't measure $\sigma_\rho^2$ for the NPs constituent atoms during imaging, but we can assume that $\sigma_\rho^2$ is identical for all atoms. Therefore, the relative variation of the atomic intensity distributions between atoms is a measure of relative mean-square atomic displacements between atoms. In this study, we denoted the isotropic variance of the atomic intensity distribution as $U_{iso} = \sigma_g^2$. We fitted the intensity distributions of the atoms in

the reconstructed 3D volume with isotropic 3D Gaussians using a least square's approach to measure $U_{iso}$. We used the following objective function to make the fitting problem linear:

$$S_{iso} = \sum_{i=0}^{n} I_i \Big( \ln(I_i) - (\beta_0 + \beta_1 r_i^2) \Big)^2$$

where $I_i$ is the intensity of the $i$th voxel in the fitting region, $(\beta_0, \beta_1)$ are the fitting parameters, and $S_{iso}$ is the quantity to be minimized. The weighting $I_i$ compensates for the fact that minimizing the sum of the square error of the natural logarithms overvalues voxels with very small intensities. We conducted fitting over a sphere of radius $\frac{3}{4} \cdot \frac{\bar{a}}{2\sqrt{2}}$, where $\frac{\bar{a}}{2\sqrt{2}}$ is half the distance between nearest neighboring atoms in an FCC lattice and the factor of $\frac{3}{4}$ prevented fitting of the tails of other atoms' intensity distributions.

*Anisotropic atomic displacement.* The atomic displacements are not necessarily isotropic (identical in all directions). To account for anisotropy of the atomic displacements, the Debye–Waller factor can be generalized to a trivariate Gaussian distribution:

$$T(\boldsymbol{q}) = \exp[-2\pi \boldsymbol{q}^T \mathbf{U} \boldsymbol{q}]$$

where $\mathbf{U}$ is the $3 \times 3$ anisotropic displacement parameter matrix with elements

$$\mathbf{U} = \begin{bmatrix} \langle x^2 \rangle & \rho_{xy}\langle x \rangle\langle y \rangle & \rho_{xz}\langle x \rangle\langle z \rangle \\ \rho_{xy}\langle x \rangle\langle y \rangle & \langle y^2 \rangle & \rho_{yz}\langle y \rangle\langle z \rangle \\ \rho_{xz}\langle x \rangle\langle z \rangle & \rho_{yz}\langle y \rangle\langle z \rangle & \langle z^2 \rangle \end{bmatrix}$$

denoting the correlation between displacements in the $x$ and $y$ directions as $\rho_{xy}$ and so forth (not to be confused with the charge density, also denoted as $\rho$). Since the atomic scattering factor is well approximated by an isotropic Gaussian that is identical across atoms, the observed real-space atomic intensity distributions can be modeled as a trivariate Gaussian distribution

$$g(\mathbf{r}) \propto \exp\left[ -\frac{1}{2}\mathbf{r}^T \boldsymbol{\Sigma}^{-1} \mathbf{r} \right]$$

where $\boldsymbol{\Sigma}$ is the $3 \times 3$ symmetric covariance matrix. The covariance matrix can be expressed as $\boldsymbol{\Sigma} = \mathbf{PDP}^{-1}$, where $\mathbf{D}$ is a $3 \times 3$ diagonal matrix with elements equal to the eigenvalues of $\boldsymbol{\Sigma}$ and $\mathbf{P}$ is a $3 \times 3$ matrix whose columns are the corresponding eigenvectors of $\boldsymbol{\Sigma}$. Denoting the eigenvalues of $\boldsymbol{\Sigma}$ as $\sigma_{g,1}, \sigma_{g,2}, \sigma_{g,3}$ in descending order and the corresponding normalized eigenvectors as $\hat{\boldsymbol{v}}_1, \hat{\boldsymbol{v}}_2, \hat{\boldsymbol{v}}_3$ then $g(\mathbf{r})$ can be expressed in the principal coordinate system as

$$g(\mathbf{r}) \propto e^{\frac{-v_1^2}{2\sigma_{g,1}^2}} e^{\frac{-v_2^2}{2\sigma_{g,2}^2}} e^{\frac{-v_3^2}{2\sigma_{g,3}^2}}$$

where $\mathbf{r} = v_1\hat{\boldsymbol{v}}_1 + v_2\hat{\boldsymbol{v}}_2 + v_3\hat{\boldsymbol{v}}_3$ in the new coordinate system. The anisotropic displacement matrix $\mathbf{U}$ has the same eigenvectors as $\boldsymbol{\Sigma}$, and so, using the same techniques as in the isotropic case, we recover the anisotropic variance relations

$$U_{maj} = \sigma_{g,1}^2 = \sigma_\rho^2 + \langle u_{v_1}^2 \rangle$$

$$U_{med} = \sigma_{g,2}^2 = \sigma_\rho^2 + \langle u_{v_2}^2 \rangle$$

$$U_{min} = \sigma_{g,3}^2 = \sigma_\rho^2 + \langle u_{v_3}^2 \rangle$$

Since $\sigma_\rho^2$ is unknown, we can't measure the mean-square atomic displacements along the eigenvectors, but relative anisotropic atomic displacement parameters describe the relative mean-square atomic displacements between atoms. Furthermore,

we define an atom's degree of isotropy DOI as

$$\mathrm{DOI} = U_{min}/U_{maj}$$

An atom has a DOI of one if its atomic displacements are purely isotropic and zero if they are confined to a plane or a line. For anisotropic displacements, $\boldsymbol{\Sigma}$ was calculated using a least square's fitting procedure with the objective function

$$S_{aniso} = \sum_{i=0}^{n} I_i \Big( \ln(I_i) - (\beta_0 + \beta_1 x_i^2 + \beta_2 y_i^2 + \beta_3 z_i^2 + \beta_4 xy + \beta_5 xz + \beta_6 yz) \Big)^2$$

The inverse covariance matrix $\boldsymbol{\Sigma}^{-1}$ is calculated from the fitting parameters $(\beta_0, ..., \beta_6)$ as

$$\boldsymbol{\Sigma}^{-1} = \begin{bmatrix} -2\beta_1 & -\beta_4 & -\beta_5 \\ -\beta_4 & -2\beta_2 & -\beta_6 \\ -\beta_5 & -\beta_6 & -2\beta_3 \end{bmatrix}$$

As described above, $U_{maj}$, $U_{med}$, $U_{min}$ are equal to the square roots of the eigenvalues of $\boldsymbol{\Sigma}$. The entire $\mathbf{U}$ matrix can be recovered by transforming back into the $(x,y,z)$ coordinate system.

Anisotropic displacement measurements are ignored in the rare case that one of $U_{maj}$, $U_{med}$, $U_{min}$ are negative or 4 times the square of the fitting radius, indicating an intensity distribution that can't be modeled with a multivariate Gaussian. For the eight NPs analyzed, anisotropic displacement calculations were ignored for one atom in NP1 and one atom in NP3.

## Data availability
The data that support the findings of this study are available from the corresponding author upon reasonable request.

## Code availability
Our code developments are available open source as part of the SINGLE software package, available for download at https://github.com/hael/SIMPLE3.0.git.

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

## Acknowledgements

H.W., R.M.-P., C.F.R., C.T.S.V., and H.E. were supported by the Intramural Research Program of the NIH. J.P. acknowledges Institutes for Basic Science (IBS-R006-D1), the National Research Foundation of Korea (NRF) grant funded by the Korean government (MSIT) (No. NRF-2017R1A5 A1015365, and No. NRF-2020R1A2C2101871). S.K., J.H., and J.P. acknowledge support by Samsung Science and Technology Foundation under project number SSTF-BA1802-08 for the development of the reconstruction algorithm. B.H.K. acknowledges the National Research Foundation of Korea (NRF) (NRF-2021R1C1C1014339). The experiments were performed at the Molecular Foundry, Lawrence Berkeley National Laboratory, which is supported by the U.S. Department of Energy under contract no. DE-AC02-05CH11231.

## Author contributions

Conception/design of the work: H.W., C.M., C.F.R., J.H., P.E., J.P., H.E. Software design: H.W., R.M.-P., C.M., C.F.R., C.T.S.V., H.E. All authors, including S.K., B.H.K. and S.K., contributed to acquisition/analysis/interpretation of data and writing of the manuscript.

## Funding

## Competing interests

The authors declare no competing interests.
