## [Peer Review File · Communications Chemistry]

Reviewers' comments:

Reviewer #1 (Remarks to the Author):

This manuscript introduces a computational procedure to study the atomic mobility of Pt nanoparticles using atomic resolution movie. Through the analysis the authors concluded that small, solubilized platinum nanocrystals have a structure of the ordered core surrounded by mobile surface atoms. The codes developments are available open source online, and will no doubt benefit the electron microscopy community in terms of data processing. I recommend publication after the authors address the following questions:

1. The manuscript lacks the discussion of the beam effect on the main conclusion that Small (< 3 nm), solubilized Platinum nanocrystals—an ordered core surrounded by mobile surface atoms. While the authors acknowledge that "crystal abnormalities" could result from NP surface interactions with ligands, the solvent, and the electron beam", no analysis or quantification is given to show whether the main conclusion will still be valid without the electron beam. Considering the amount of electron is needed to irradiate the sample to acquire an atomic resolution movie, the ligand and solution will undergo significant chemical change with non-negligible concentration of different types of radicals, it's possible that the observed mobility of the outer shell atoms are beam induced. Dose controlled experiments could address this uncertainty.

2. The author states that the procedure provides basis for an automatic and unbiased process. In the denoising step of the computation method, prior knowledge of the pt nanoparticle structure is needed. The author also mentioned that "We knew from our previous work that the solution ensemble of synthesized Pt NPs is heterogenous—each NP has a unique atomic structure". This means the structure of the particle being analyze needs to be predetermined thus could introduce bias. Additional clarification is needed to understand why the procedure is unbiased.

Minor comments:

3. There are plenty of texts being cut off in some of the figures, such as in Figure 1, Figure 3, and Figure 4

4. How is the center of mass determined? Please provide definitions or references. It is not clear if it truly represents the location with the highest probability of finding the atom.

5. Line 308, the author states that "An initial threshold T_0 was estimated using Otsu's method and refined as follows. We produced the grey-level histogram of map C, and generated 15 uniform thresholds in the range from T_0 to the maximum value of map C." Why use the estimated threshold T_0 as the minimum value of the optimal threshold? One would assume an estimation could be off in both directions.

Reviewer #2 (Remarks to the Author):

In the present manuscript the authors show the in-situ structural properties of the surface atoms in Pt nanoparticles. They show that the atoms on the outer layers of the nanoparticles show a certain mobility and thermal motion, which might modify their coordination number. The authors use an accurate unsupervised statistical analysis, which allows to study the nanoparticles at the atomic scale, from core to shell. The core is found to have an ordered structure, while the shell is clearly unstable with high mobility atoms. Studying the surface dynamics is of high importance in order to understand the chemical properties of this type of nanostructures. In fact, the present work will help understanding the surface catalytic reactions of interest for energy and environmental applications.

I see a great potential for the methodology, specially taking into account that the accurate measurement of the coordination number/state can help to determine the final catalytic activity of any catalyst. This methodology is adapted from single particle cryo-TEM methodologies, applied to small proteins. In the case of proteins, it can be applied thanks to the reproducibility and high repeatability of the structures from particle to particle.

I suggest publication after discussing the following issue:

I would like to ask the authors to discuss what would happen in case the morphology of the nanoparticles to be studied is differing a bit more in size, number of atoms and atomic distribution. Could this methodology be adapted?

Reviewer #1

This manuscript introduces a computational procedure to study the atomic mobility of Pt nanoparticles using atomic resolution movie. Through the analysis the authors concluded that small, solubilized platinum nanocrystals have a structure of the ordered core surrounded by mobile surface atoms. The codes developments are available open source online, and will no doubt benefit the electron microscopy community in terms of data processing. I recommend publication after the authors address the following questions:

1. The manuscript lacks the discussion of the beam effect on the main conclusion that Small (< 3 nm), solubilized Platinum nanocrystals—an ordered core surrounded by mobile surface atoms. While the authors acknowledge that "crystal abnormalities" could result from NP surface interactions with ligands, the solvent, and the electron beam", no analysis or quantification is given to show whether the main conclusion will still be valid without the electron beam.

Response: *We have included analysis of the beam effect on the main conclusion.*

Considering the amount of electron is needed to irradiate the sample to acquire an atomic resolution movie, the ligand and solution will undergo significant chemical change with non-negligible concentration of different types of radicals, it's possible that the observed mobility of the outer shell atoms are beam induced. Dose controlled experiments could address this uncertainty.

Response: *We have included analysis of the beam effect and updated the discussion with a suggestion for a suitable dose-controlled experiment.*

2. The author states that the procedure provides basis for an automatic and unbiased process. In the denoising step of the computation method, prior knowledge of the pt nanoparticle structure is needed. The author also mentioned that "We knew from our previous work that the solution ensemble of synthesized Pt NPs is heterogenous—each NP has a unique atomic structure". This means the structure of the particle being analyze needs to be predetermined thus could introduce bias. Additional clarification is needed to understand why the procedure is unbiased.

Response: *We have removed all occurrences of "unbiased" in the manuscript and clarified the language. Of course, 3D reconstructions need to be generated prior to the analysis, but this can be done in an unbiased fashion using SINGLE.*

Minor comments:

3. There are plenty of texts being cut off in some of the figures, such as in Figure 1, Figure 3, and Figure 4

Response: *We have corrected this.*

4. How is the center of mass determined? Please provide definitions or references. It is not clear if it truly represents the location with the highest probability of finding the atom.

Response: We have included a reference. It truly represents the location with the highest probability of finding the atom.

5. Line 308, the author states that “An initial threshold T0 was estimated using Otsu’s method and refined as follows. We produced the grey-level histogram of map C, and generated 15 uniform thresholds in the range from T0 to the maximum value of map C.” Why use the estimated threshold T0 as the minimum value of the optimal threshold? One would assume an estimation could be off in both directions.

Response: No. The bias is always unidirectional.

Reviewer #2

In the present manuscript the authors show the in-situ structural properties of the surface atoms in Pt nanoparticles. They show that the atoms on the outer layers of the nanoparticles show a certain mobility and thermal motion, which might modify their coordination number. The authors use an accurate unsupervised statistical analysis, which allows to study the nanoparticles at the atomic scale, from core to shell. The core is found to have an ordered structure, while the shell is clearly unstable with high mobility atoms. Studying the surface dynamics is of high importance in order to understand the chemical properties of this type of nanostructures. In fact, the present work will help understanding the surface catalytic reactions of interest for energy and environmental applications.

I see a great potential for the methodology, specially taking into account that the accurate measurement of the coordination number/state can help to determine the final catalytic activity of any catalyst. This methodology is adapted from single particle cryo-TEM methodologies, applied to small proteins. In the case of proteins, it can be applied thanks to the reproducibility and high repeatability of the structures from particle to particle.

I suggest publication after discussing the following issue:

I would like to ask the authors to discuss what would happen in case the morphology of the nanoparticles to be studied is differing a bit more in size, number of atoms and atomic distribution. Could this methodology be adapted?

Response: Yes. This has been expensively described in our previous publications, listed below.

Correlating 3D Surface Atomic Structure and Catalytic Activities of Pt Nanocrystals

S. Kim, J. Kwag, C. Machello, S. Kang, J. Heo, C. F. Reboul, et al.

Nano Lett 2021 Vol. 21 Issue 2 Pages 1175-1183

Critical differences in 3D atomic structure of individual ligand-protected nanocrystals in solution

B. H. Kim, J. Heo, S. Kim, C. F. Reboul, H. Chun, D. Kang, et al.

Science 2020 Vol. 368 Issue 6486 Pages 60-67

Method for 3D atomic structure determination of multi-element nanoparticles with graphene liquid-cell TEM

J. Heo, D. Kim, H. Choi, S. Kim, H. Chun, C. F. Reboul, et al.

Sci Rep 2023 *Vol. 13 Issue 1 Pages 1814*

Nanoparticle imaging. 3D structure of individual nanocrystals in solution by electron microscopy

J. Park, H. Elmlund, P. Ercius, J. M. Yuk, D. T. Limmer, Q. Chen, et al.

Science 2015 *Vol. 349 Issue 6245 Pages 290-5*

SINGLE: Atomic-resolution structure identification of nanocrystals by graphene liquid cell EM

C. F. Reboul, J. Heo, C. Machello, S. Kiesewetter, B. H. Kim, S. Kim, et al.

Sci Adv 2021 *Vol. 7 Issue 5*

REVIEWERS' COMMENTS:

Reviewer #1 (Remarks to the Author):

I think the authors have addressed all my concerns with their best knowledge and I recommend publication at this stage.